# Unsteady Flow Characteristics of Rotating Stall and Surging in a Backward Centrifugal Fan at Low Flow-Rate Conditions

**Biao Zhou [1], Ximing He [2], Hui Yang [1,3,*,†], Zuchao Zhu [1], Yikun Wei [1,3] and Yan Zhang [4,*,†]**

1 National-Provincial Joint Engineering Laboratory for Fluid Transmission System Technology, Zhejiang Sci-Tech University, Hangzhou 310018, China; zb18395588895@163.com (B.Z.); zhuzuchao@zstu.edu.cn (Z.Z.); yikunwei@zstu.edu.cn (Y.W.)
2 Beijing Aerospace Control Center, Beijing 100094, China; hxm_at1991@163.com
3 Shangyu Industrial Technology Research Institute, Zhejiang Sci-Tech University, Shaoxing 312000, China
4 Department of Aeronautics, Imperial College London, London SW7 2AZ, UK
* Correspondence: yanghui@zstu.edu.cn (H.Y.); yan.zhang1@imperial.ac.uk (Y.Z.)
† These authors contributed equally to this work.

**Abstract:** The steady and unsteady flow characteristics of internal flow in a backward centrifugal fan of double inlet at low flow-rate condition are investigated by computational fluid dynamics in this paper. The investigation aims to reveal insights into generation mechanisms and our physical understanding of the rotating stall and surge. The numerical results mainly demonstrate that, with decreasing flow rate, a large number of vortex flows almost increasingly occupy the internal flow of the impeller. The reverse flow and separation vortices increasingly appear near the outlet of volute, and the internal flow of the impeller is completely blocked by the separated vortex flow at low flow-rate conditions. Results indicate that, due to a synchronization of the impeller rotation and separation vortex, these separated vortices act intensely on the pressure surface of the blade with time evolution, and the interaction between the separated vortices and surface of blade increasingly yields small-scale eddies. It is further found that the amplitude of pressure and velocity fluctuations gradually increase with the decrease of flow rate in a certain range. The unsteady characteristics acting on the volute tongue gradually increase in a range of $Q_d$ to $0.3\,Q_d$ ($Q_d$ is the design volume flow rate) with the decrease of flow rate, and the unsteady characteristics acting on the volute tongue are weakened at the working condition of $0.15\,Q_d$. These insights clearly explain the unsteady nature of the rotating stall and surge phenomenon in the double inlet backward centrifugal fan.

**Keywords:** centrifugal fan; large eddy simulation; pressure fluctuation; surge; unsteady flow

## 1. Introduction

Centrifugal fans are important equipment broadly used in a variety of civil and industrial fields [1,2]. With the rapid development of the economy, the global energy shortage is significant. Human's living environment has been facing more and more serious pollution, and sustainable development and environmental protection have become basic national policies in the world [3]. The double-suction centrifugal fan has large flow coefficient, high operating reliability, safety and stability, and the air enters the double-suction fan from both sides of the impeller. With the rapid development of various industries, it is not only required to provide a higher flow rate, but also to put forward higher and higher requirements on the miniaturization and efficiency of fans. Therefore, it is urgent to study the internal flow patterns of double-suction centrifugal fans. The study of internal flow law and unsteady characteristics of centrifugal fan blade wake and suction surface separation

flow under the condition of small flow rate involves undertaking basic research and preparation for improving the model. The prediction of flow patterns in centrifugal fans has been considered a difficult problem by turbomachinery researchers and designers. It is important to study the internal flow rules of double-suction centrifugal fans. Fike et al. [4] visualized the flow field in the rotor blade passage of an axial flow fan operating under rotating stall conditions using Particle Image Velocimetry (PIV). The results show that the generation and development of flow distortion are obviously influenced by the advance of the rotating stall cell and the recovery of normal flow with the movement of the rotating stall cell. Norimasa et al. [5] used a hot-wire anemometer to study the rotor inlet and the outside of the rotor tip in a semi-open propeller fan. Based on the results of velocity pulsation and power spectral density distribution, the flow field outside the rotor inlet and rotor cotyledon tip is discussed, and we also consider the blade stall factors. Xiao et al. [6] conducted a numerical simulation of the three-dimensional turbulent flow field in a centrifugal fan. The conclusion shows that, with the decrease of flow rate, the instability zone in the blade passage moves from the outlet to the inlet of the impeller, and the instability zone in different passages develops in the opposite direction to the rotation direction of the impeller. In order to deal with the urgent problems related to turbomachinery design, Day [7] summarized the research on rotating stall and flow disturbance. It is suggested that the future development of Computational Fluid Dynamics (CFD) will take the form of better performance prediction, better flow model and better interpretation of experimental results. Semlitsch and Mihaescu [8] analyzed the flow evolution through unsteady three-dimensional flow simulation of centrifugal compressor. Comparing the design condition with the low flow condition with surge, it is found that the tip leakage phenomenon appears around the impeller under the design condition. The reversed flow presents swirling motion in the impeller, which affects the incidence angle of the blade, then affects the efficiency. By means of proper orthogonal decomposition and dynamic modal decomposition, the flow structure with surge is analyzed comprehensively. Yang and Meng [9] performed unsteady numerical simulations of the flow field in the centrifugal fan to determine the causes of vibration and noise. The results show that the interaction between the flow of non-uniform impeller and the fixed volute causes significant pressure fluctuation, which is an important source of vibration and noise of centrifugal machinery. In order to optimize the performance, Ding [10] et al. used CFD to analyze the internal flow of a certain type of three-stage fan at the rated speed under different total pressures. The causes of eddy current and backflow in the fan are analyzed, and the methods to improve the efficiency of the fan are given. Sushanlal and Anish [11] performed steady and unsteady simulations using the Reynolds Averaged Navier Stokes equations and the Shear Stress Transfer (SST) turbulence model. In this paper, the vortex structure of steady and unsteady blade wakes is analyzed in detail, and the interaction between jet purge flow and mainstream flow in the presence of upstream disturbance/wakes is studied with numerical investigation. Abramian and Howard [12] and Ubaldi et al. [13] made use of a laser Doppler speedometer (LDV) to monitor the impeller and semi-open impeller of low speed centrifugal fan in detail. Bianchi et al. [14] argued that a rotating stall is not a single phenomenon, but rather four obvious phenomena of part span, full span, small scale and large scale. Lin and Huang [15] combined fan design, prototype production, experimental verification and numerical simulation, and compared the experimental results with the numerical calculation results. The surge of fan is a self-excited cyclically nonlinear phenomenon [16], which results in the systematically mechanical damage. At least four different surge categories are distinguished with respect to unsteady flow and pressure fluctuations [17]. Researchers are in agreement that the rotating stall does lead to mechanical damage and significant failure [18–20].

In general, the airflow in centrifugal fan is inhaled axially through the air inlet on both sides, and then enters the impeller channel through a 90° bend. The energy is obtained through the rotating impeller, and finally discharged from the exhaust port through the concentration and diversion effect of the volute. However, once the flow rate continues to gradually decrease, the pressure of centrifugal fan begins to drop sharply [21]. Due to the generation of separation vortex in the impeller, the blade paths of the centrifugal fan impeller at low flow-rate conditions will be extremely easy to fill with a

large number of turbulent structures, which leads to the efficiency of centrifugal fan dropping rapidly, causing the blade channel to be blocked and even resulting in a rotating stall and surge. A large number of passive and active flow control strategies have been employed to extend the operation scope of the centrifugal fan. The studies of the rotating stall and surge is in debate, and the community has comprehensively focused on the rotating stall in single stage axial and centrifugal compressors; there is disagreement regarding the importance of rotating stall [22]. A better understanding the large-scale flow structures involves inventing improved techniques for delaying the rotating stall and surge. Once the internal flow unsteady characteristics of centrifugal fan at low flow-rate conditions have been described in detail, the rotating stall and surge will be precisely understood.

Based on the above discussions, complex unsteady characteristics caused by separation vortex in the narrow blade passage of the impeller plays a dominant role in the rotating stall and surge, and sensitively affects the efficiency of the centrifugal fan. Thus, it is essential to reveal the unsteady characteristics of a large number of turbulent structures in the centrifugal fan due to low flow-rate conditions. Our study mainly focuses on the steady flow and the unsteady flow characteristics about the complex flow of the impeller and volute with decreasing flow rate. The characteristics mainly include velocity, pressure and secondary flow distribution, and separation flow distribution. Based on these characteristics, this paper attempts to explain the mechanism of the emergence and evolution of the flow phenomenon. The origins and effects of decreasing flow rate on the fluctuations of pressure and velocity, the fluctuations spectral analysis of pressure and velocity are discussed, and we try to understand the rotating stall and surge physical phenomenon of centrifugal fan. The results are of physical and mechanical significance for the low-speed centrifugal fans widely used in ventilation circumstances under the conditions and parameters that are as discussed in this paper, which are quite typical in engineering applications such as indoor and basement ventilation. Thus, our results will provide meaningful guidance in related applications.

## 2. Computational Geometric Model and Parameter of Backward Centrifugal Fan

In this section, a backward centrifugal fan and computational geometric model will be introduced. Figure 1 shows the geometric model of the fluid domain. As shown in Figure 1, one can see that the computational domain of centrifugal fan model consists of two lengthened inlet domains, the impeller domain, a volute domain and a lengthened outlet domain to obtain its performance and simulate its internal unsteady flow characteristics. In this paper, the centrifugal fan with a double-impeller with 12 backward blades are studied by numerical simulations. Figure 2 describes the geometric model of the impeller and volute. In Figure 2, we can see that the centrifugal fan mainly includes the left impeller, right impeller, a volute and an axis. The design condition of this centrifugal fan is 1400 rpm and 12,000 $m^3$/h. The geometric key parameters of the model are given in Table 1.

**Table 1.** Geometric parameter for the centrifugal fan.

| Parameter | Dimension |
|---|---|
| Blade inner diameter (d1) | 368 mm |
| Blade outer diameter (d2) | 500 mm |
| Blade width (b) | 126 mm |
| Blade length (I) | 150 mm |
| Blade inlet angle ($\beta_1$) | 19.5° |
| Blade outlet angle ($\beta_2$) | 47° |
| Number of blades (Z) | 12 |
| Volute width (B) | 668 mm |

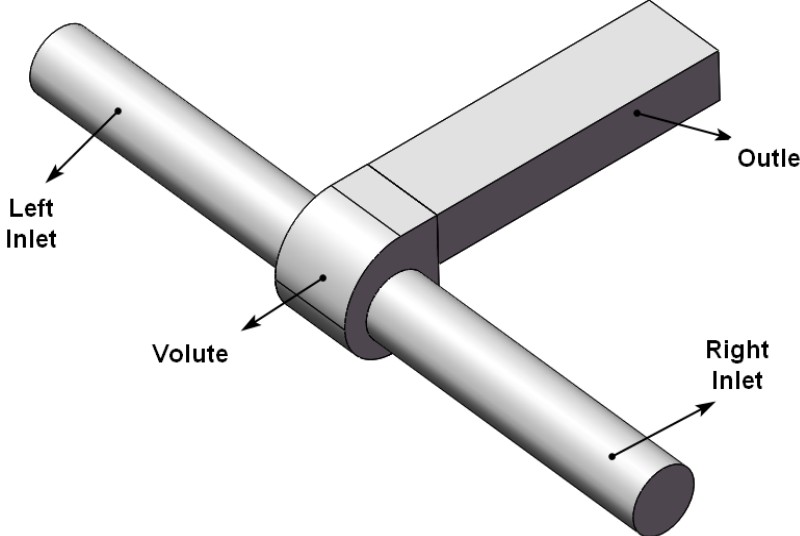

**Figure 1.** Geometric model of fluid domain.

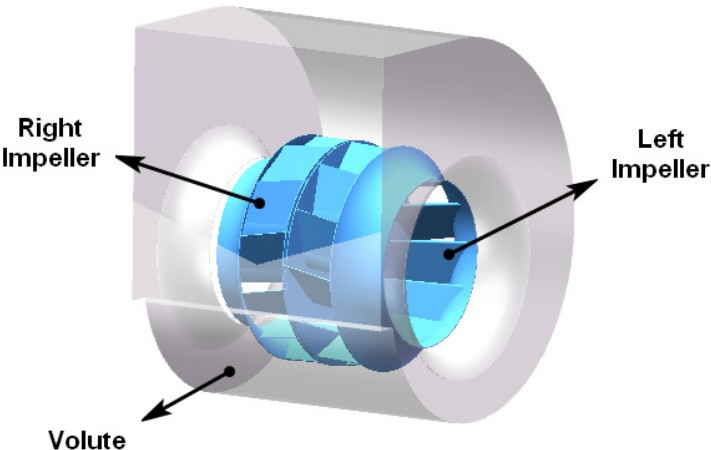

**Figure 2.** Geometric model of impeller and volute.

## 3. Numerical Methods and Experimental Test

In this section, the governing equation, turbulent model, verification of grid independence, and computational boundary condition are introduced. In this paper, finite volume method is used as the solver of the governing equations. Fluent is applied to set the turbulence model, boundary conditions and solving methods.

### 3.1. Governing Equations and Turbulence Model

CFD is an effective tool to study the physical mechanism of fluid mechanics [23–28]. The continuity equation is based on the law of conservation of mass [29]:

$$\frac{\partial u}{\partial x} + \frac{\partial v}{\partial y} + \frac{\partial w}{\partial z} = 0 \tag{1}$$

The momentum equation is based on Newton's second law, and its form is:

$$\frac{du_i}{dt} = \frac{\partial u_i}{\partial t} + u_j \frac{\partial u_i}{\partial x_j} + u_k \frac{\partial u_i}{\partial x_k} = f_i - \frac{1}{\rho} \frac{\partial p}{\partial x_i} + \nu \nabla^2 u_i \tag{2}$$

The re-normalization group (RNG) k-e turbulence model is used for steady simulation, and the large eddy simulation (LES) turbulence model is used for unsteady simulation [30,31]. Turbulence is inevitable in the centrifugal fan. Turbulence is a highly non-linear and complex flow. The turbulence kinetic energy equation and diffusion equation of RNG k-$\varepsilon$ turbulence model are [32]:

$$\frac{\partial}{\partial t}(\rho k) + \frac{\partial}{\partial x_i}(\rho k u_i)_k = \frac{\partial}{\partial x_j}\left(\alpha_k \mu_{eff} \frac{\partial k}{\partial x_j}\right) + G_k + G_b - \rho\varepsilon - Y_M + S_k \tag{3}$$

$$\frac{\partial}{\partial t}(\rho\varepsilon) + \frac{\partial}{\partial x_i}(\rho\varepsilon u_i)_\varepsilon = \frac{\partial}{\partial x_j}\left(\alpha_\varepsilon \mu_{eff} \frac{\partial\varepsilon}{\partial x_j}\right) + C_{1\varepsilon}\frac{\varepsilon}{k}(G_k + G_{3\varepsilon}G_b) - C_{2\varepsilon}\rho\frac{\varepsilon^2}{k} - R_\varepsilon + S_\varepsilon \tag{4}$$

where $G_k$ represents the turbulent kinetic energy generated by the velocity gradient of the laminar flow, $G_b$ represents the turbulent kinetic energy caused by buoyancy, $Y_M$ represents the wave generated by the transition diffusion in the compressible turbulent flow, C is a constant, the turbulent Prandtl Numbers of k equation and e equation respectively, and $S_k$ and $S_\varepsilon$ are defined by the user.

The LES approach had been used to capture the unsteady flow-field at off-design operating conditions. According to the basic idea of LES, an average method must be adopted to distinguish large scale vortices that can be solved and small-scale vortices to be modeled [33]. Let variable $u_i$ be decomposed into $\overline{u}_i$ and subgrid variables (modeled variables) $u'_i$. That is $u_i = \overline{u}_i + u'_i \cdot \overline{u}_i$ is expressed in the formula proposed by Leonard:

$$\overline{u}_i(x) = \int_{-\infty}^{+\infty} G(x - x')u_i(x')dx' \tag{5}$$

where $G(x - x')$ denotes the filter function. By applying the filter function to the terms of the Navier-Stokes equation, the filtered turbulence control equation can be obtained:

$$\frac{\partial\overline{u}_i}{\partial t} + \overline{u}_j\frac{\partial\overline{u}_i}{\partial x_j} + \overline{u}_k\frac{\partial\overline{u}_i}{\partial x_k} = f_i - \frac{1}{\rho}\frac{\partial p}{\partial x_i} + v\nabla^2\overline{u}_i - \frac{\partial\tau_{ij}}{\partial x_j} - \frac{\partial\tau_{ik}}{\partial x_k} \tag{6}$$

where, $\tau_{ij}$ and $\tau_{ik}$ represent the influence of small vortices on large vortices. In this paper, the second order upwind scheme is adopted, and the upwind difference scheme takes full account of the influence of flow direction on the difference formula, which makes the calculation results more accurate.

### 3.2. Verification of Grid Independence

In order to accurately capture the complex flow properties, the structured grid and unstructured grid are implemented in the whole computational domain, where structured grids are produced in the inlet and outlet sections, and the impeller and the volute and other complex structures are divided by unstructured grids. The mesh details of the volute are shown in Figure 3.

In order to determine the final grid implemented in the following numerical simulation, four groups of grids are used to verify the grid independence verification. The number of grids in these four groups is successively doubled and all numerical simulations are conducted at the same conditions. The relationship between the number of grids and the total pressure of the fan is described in Figure 4. It can be seen that when the number of grids is greater than $2 \times 10^6$, the results of simulation calculation can achieve the ideal convergence effect. Table 2 shows the final grid scheme. In Table 2, all grid numbers in all computational domains are presented.

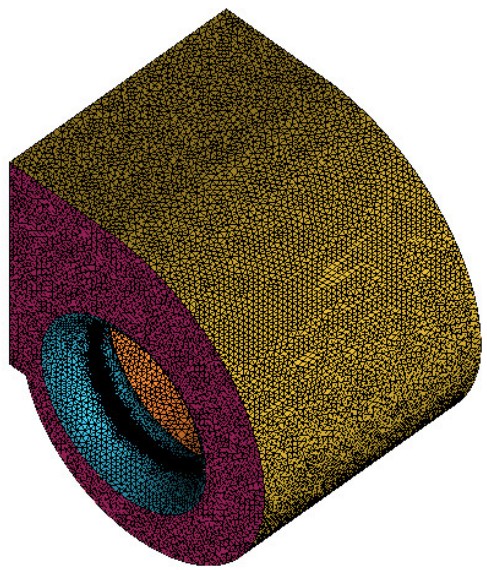

**Figure 3.** Mesh generation.

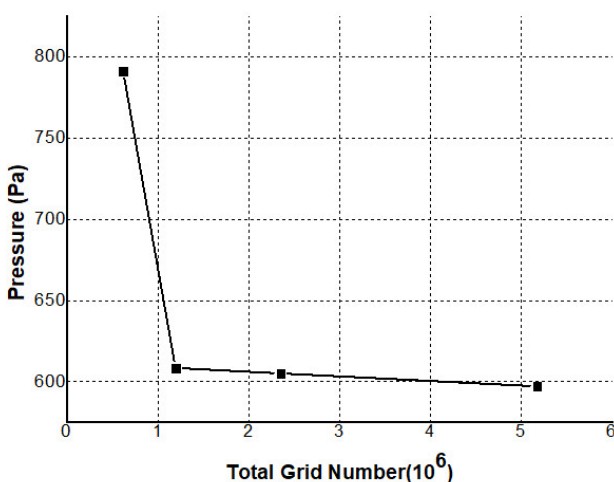

**Figure 4.** Grid dependence test.

**Table 2.** Number of grids of the computational domain.

| Component | Number of Grids/$10^3$ |
|---|---|
| Left Impeller domain | 1109 |
| Right Impeller domain | 1237 |
| Left Inlet duct domain | 218 |
| Right Inlet duct domain | 218 |
| Outlet duct domain | 67 |
| Volute domain | 414 |
| Total | 3263 |

### 3.3. Boundary Conditions

In this subsection, the boundary conditions of all numerical simulations are introduced, where the density of air is 1.225 kg/m$^3$ and the viscosity is $1.7894 \times 10^{-5}$ kg/(m·s). The surfaces of all impeller blades, hub and motor are defined as the rotating wall surfaces, and the rotation speed is given as 1400 r/min. The inner surface of the volute and the inner surface of the air duct are stationary walls. The no slip interface is installed between the rotating wall and the stationary wall. The static pressure at the outlet is consistent with the atmospheric pressure. The air inlet of the fan is the inlet of the air

ducts on the left and right sides, and the total pressure at the inlet is atmospheric pressure, which is perpendicular to the inlet surface and enters the drainage basin. The outlet is the exit of the fluid in the entire computational domain, and the outlet is set as the out-flow boundary condition.

In this paper, the shaft power is measured by the torque method, which is given by the following formula:

$$N = T \cdot n \tag{7}$$

where N is shaft power (kW), T denotes the torque of the centrifugal fan (N·m), N represents the speed of the centrifugal fan (r/min). The formula for calculating the total effective power of the centrifugal fan ($N_e$) is given as follows:

$$N_e = \frac{TP \cdot Q}{1000} \tag{8}$$

in which P is the total pressure difference of the centrifugal fan (Pa), and Q denotes the flow rate of the centrifugal fan (m$^3$/s). The formula for calculating the total pressure efficiency of the centrifugal fan ($\eta$) is as follows:

$$\eta = \frac{N_e}{N} = \frac{TP \cdot Q}{1000N} \tag{9}$$

*3.4. Experimental Test*

Experimental technology is one of the main research methods in fluid field [34,35], and also an important verification method for numerical calculation [36,37]. The Yilida's pneumatic testing laboratory will be introduced to test the performance of centrifugal fan.

Figure 5 illustrates the performance test installation of centrifugal fan. As illustrated in Figure 5, the centrifugal fan is installed on the inlet of measurement device. An auxiliary fan is installed on the outlet of measurement device. The multiple nozzles are used in the measurement device.

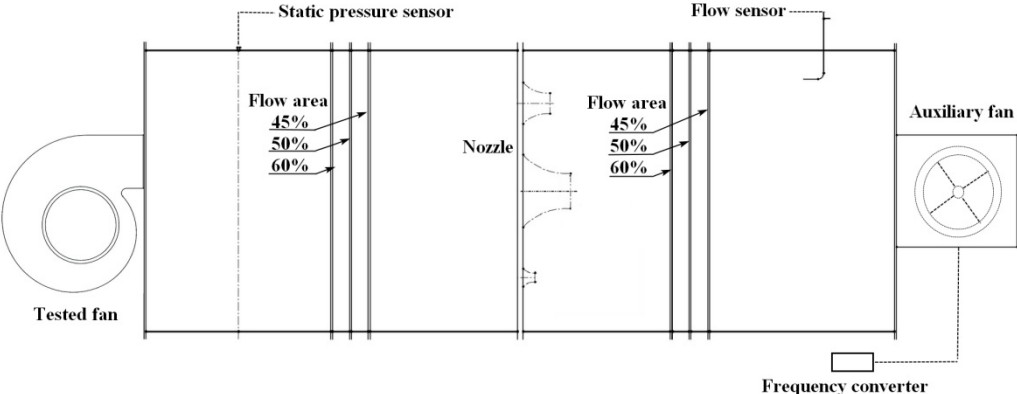

**Figure 5.** Performance test installation of centrifugal fan.

## 4. Results and Discussion

In this section, the comparison between numerical simulations and experimental test is given for aerodynamics performance of the double inlet backward centrifugal fan.

*4.1. Aerodynamics Performance of Double Inlet Backward Centrifugal Fan*

The aerodynamics performance is obtained in order to verify the accuracy of numerical simulations. The performance data of the centrifugal fan are obtained in Yilida's pneumatic testing laboratory. Figure 6 shows the comparison of aerodynamics performance between numerical simulations and experimental data at 1400 rpm. The curves of pressure-flow rates and efficiency-flow rates are illustrated in Figure 6a,b. As shown in Figure 6a, it can be seen that the flow rate increases in a certain range area, and the pressure gradually decreases. Nevertheless, the flow rate increases at small flow rate, and the pressure gradually increases. We can also obtain that the results of numerical simulations are in good

agreement with experimental results. As illustrated in Figure 6b, it is clearly observed that the pressure efficiency maintains high efficiency in a range of 10,000 to 15,000 m³/h and the results of numerical simulations are consistent with experimental results, which shows the efficiency and accuracy of the numerical computation.

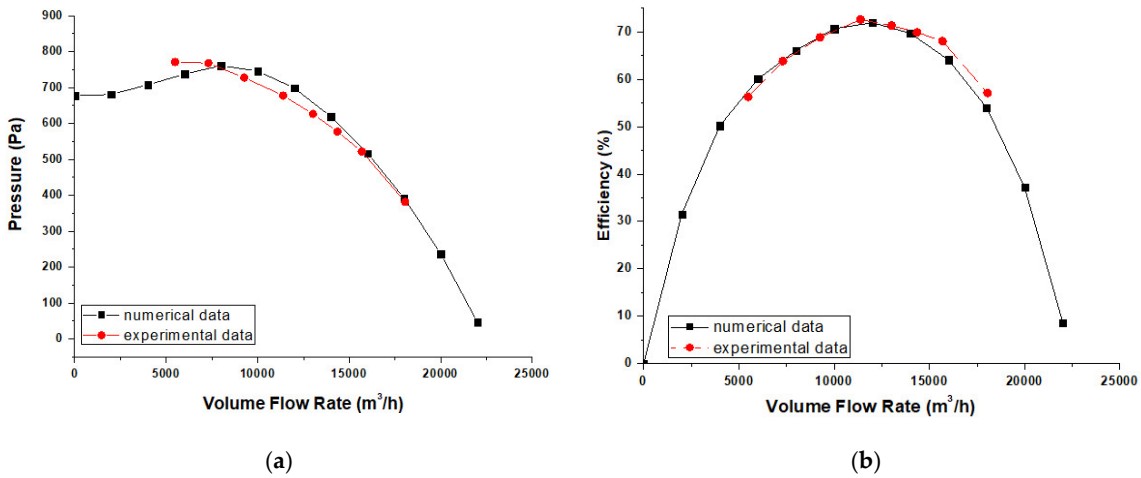

**Figure 6.** Compare of aerodynamics performance between numerical simulations and experimental data at 1400 rpm. (**a**) Pressure-Volume flow rates, (**b**) Efficiency-Volume flow rates.

*4.2. Steady Flow Field Distribution Characteristics*

4.2.1. Internal Steady Flow at Different Working Conditions

In this subsection, velocity, pressure and streamline are presented at different sections, and the distribution characteristics at the surface of revolution (z = 80 mm) and the meridional plane. Figure 7 shows the pressure contour of the rotating surface on the z = 80 mm plane. As shown in Figure 7, it is obviously seen that the low pressure occurs at the inlet of the blade, and the high gradient of pressure mainly concentrates on the blade leading and trailing edge of centrifugal fan at the designed condition of high mass flow. At 0.5 $Q_d$, the gradient of pressure increases in the blade leading and trailing edge of centrifugal fan and a higher gradient of pressure occurs in the impeller interior. At the working condition of 0.3 $Q_d$, the gradient of pressure distribution still increases and begins to be increasingly chaotic in the blade leading and trailing edge of centrifugal fan, and the average pressure force in the low-pressure zone and the high-pressure zone is lower than that of the high mass-flow zone. Meanwhile, a higher-pressure gradient region concentrates near the outlet of the volute, which indicates a large number of backflows occurring in the outlet of the volute. However, when the working condition decreases to 0.15 $Q_d$, the outlet pressure of the impeller is close to zero, and the pressure at the inlet of the impeller also clearly rises, which implies that almost no fluid flows from outlet of the volute and the delivering fluid ability of impeller has been completely destroyed. The physical phenomenon of centrifugal fan can bring about the rotating stall and surge, which can cause seriously destruction to the centrifugal fan.

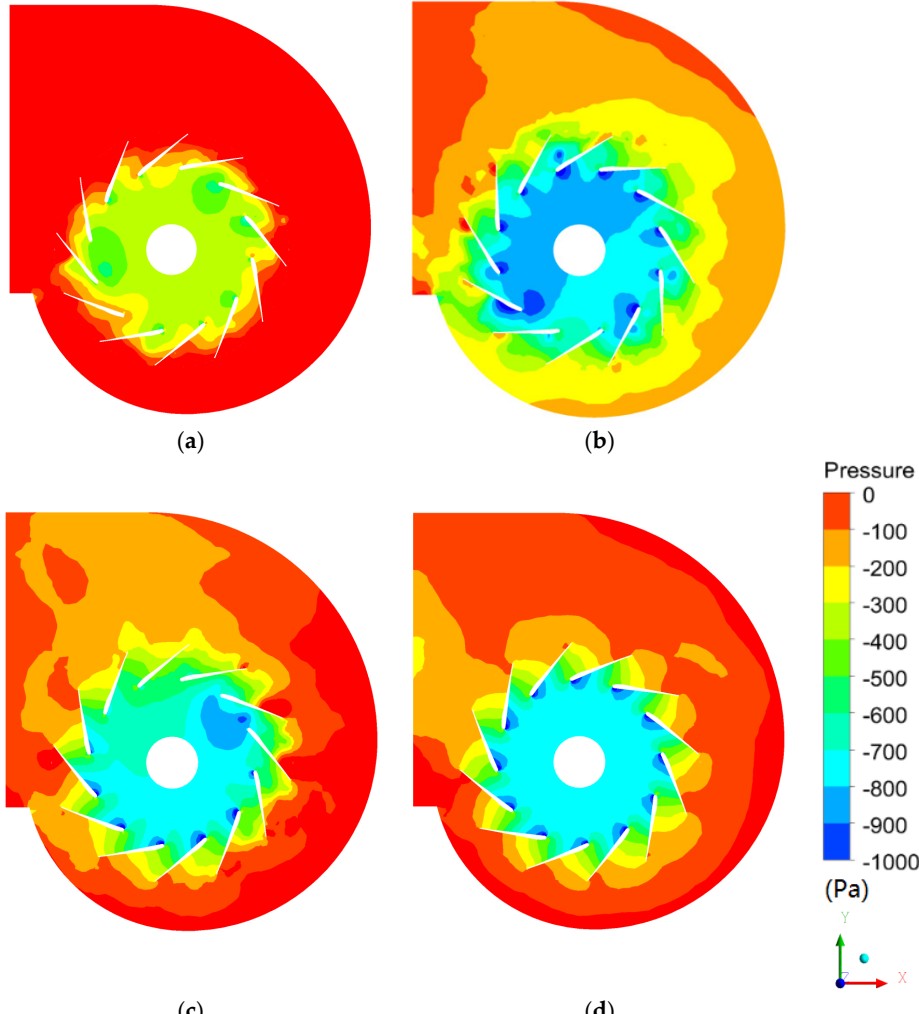

**Figure 7.** Pressure distributions of impeller and volute at z = 80 mm on the surface of revolution. (**a**) 0.15 $Q_d$, (**b**) 0.3 $Q_d$, (**c**) 0.5 $Q_d$, (**d**) $Q_d$.

Figure 8 describes the velocity distributions of impeller and volute at z = 80 mm on the surface of revolution. It is clearly observed that at designed working conditions, the high gradient of velocity mainly concentrates on the blade leading edge of impeller, the high fluid velocity occurs on the suction surface, and the low fluid velocity mainly concentrates on the pressure surface. For a better illustration of the related physical and mechanical phenomena, we use the notations of 3, 6, 9 and 12 o'clock to denote the circumferential positions on the impeller. When the mass flow rate decreases to 0.5 $Q_d$, the low-speed zone begins to appear in the blade channel from 12 o'clock to 4 o'clock, which can bring about a lot of reverse flow and turbulent vortex generation. The appearance of the low-speed area leads to the local blockage in this area and even the local reverse flow. Thus, the fluid cannot pass through the blade smoothly. At 0.3 $Q_d$, almost all the fluid in the blade passage is in a low velocity state. In the inlet area of impeller, the fluid velocity value is very great, which causes a lot of fluid to block at the inlet. The above phenomenon also indicates that the fluid cannot enter the blade channel smoothly at the inlet. Nevertheless, at 0.15 $Q_d$, the fluid velocity value in the whole basin is lower than that in the condition of high mass flow. The range of low-speed fluid in the blade channel is enlarged, especially at 12 o'clock, 4 o'clock and 8 o'clock. It is further found that a large area of low speed occurs at outlet of the volute, which can lead to a large number of backflows from the outlet of the volute.

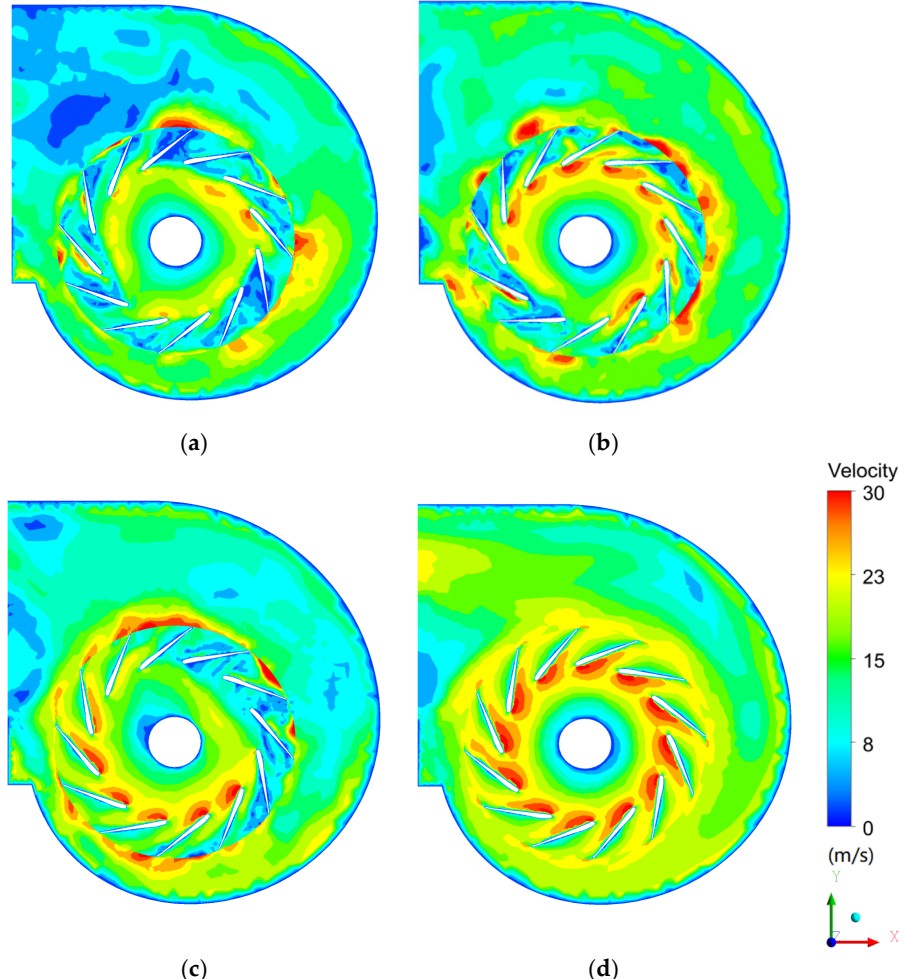

**Figure 8.** Velocity distributions of impeller and volute at z = 80 mm on the surface of revolution. (**a**) 0.15 $Q_d$, (**b**) 0.3 $Q_d$, (**c**) 0.5 $Q_d$, (**d**) $Q_d$.

Streamlines in the impeller and volute reflect the movement trend of the whole flow field and can also represent the occurrence and development of small-scale flows. Figure 9 illustrates the streamline of impeller and volute at z = 80 mm on the surface of revolution. As shown in Figure 9, we can see that the flow of the secondary flow and vortex distribution gradually increase and fill the whole impeller with a decrease of flow rate. When mass flow rate is 0.5 $Q_d$, some vortex flow and a few separation flows are generated in about one-third of the blade passage area in the impeller. In this area, the fluid in the volute even flows back into the impeller. Similarly, there is a large amount of low-speed fluid around the tongue. Since the flow rate decreases to 0.5 $Q_d$, the energy flowing in the mainstream potential flow is no longer enough to make the fluid move continuously along the boundary layer. At this point, when the fluid continues to accumulate near the outlet, the pressure in the vicinity will increase, and under the action of the reverse pressure difference, it will produce a reverse backflow upstream, crowding out the upstream fluid. In Figure 8c, at 0.5 $Q_d$, we can observe that two such separation flows occur in the direction of 12 o'clock and 10 o'clock, which will then coil the fluid of the mainstream and generate the pressure gradient perpendicular to the mainstream. This is the reason why the direction and regularity of the mainstream are destroyed. In Figure 9b, the flow rate is further reduced to 0.3 $Q_d$, and only a small amount of fluid flows into the volute in each blade passage, almost producing a separation point at the tail of each blade, which is obviously affected by the separation within the blade passage. A lot of vortex separation is mainly concentrated on the outlet part of the volute, which can result in a large-scale rotating stall. However, at 0.15 $Q_d$ in

Figure 9a, the separation zone in the blade passage almost fills all the blades, and the separation zone in the volute is also distributed around the impeller. There is only a small amount of fluid passing through the blade channel at the blade channel of 11 o'clock, 3 o'clock, and 7 o'clock. It also is observed that the outlet of these blade channels is the separation point of the flow in the volute, which leads to the flow separation phenomenon in many areas of 2 o'clock, 6 o'clock, and 10 o'clock. Under the action of surface separation at the periphery of the impeller and flow separation at the outlet position, a large number of reverse flow and separation vortices are generated near the outlet of the volute. The fluid in the outlet area of the volute is even sucked back into the volute and into the next flow cycle.

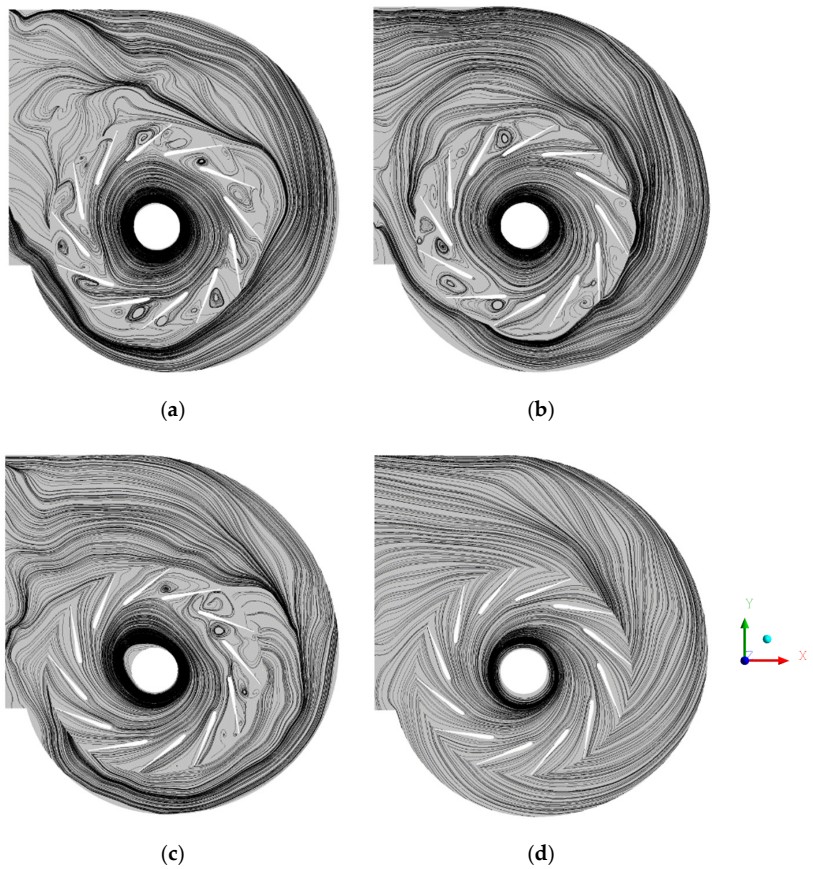

(a)      (b)

(c)      (d)

**Figure 9.** Streamline of impeller and volute at z = 80 mm on the surface of revolution. (**a**) 0.15 $Q_d$, (**b**) 0.3 $Q_d$, (**c**) 0.5 $Q_d$, (**d**) $Q_d$.

In order to further reveal the internal flow in the centrifugal fan with decreasing flow rate, the internal streamline of the only impeller is presented. Bianchi et al. [14] argued that the rotating stall is not a single phenomenon, but rather four obvious phenomena of part span, full span, small scale and large scale. Figure 10 shows the separate discussion of the impeller part in Figure 9, which is the streamline of the impeller at z = 80 mm on the surface of revolution. In Figure 10d, without any flow separation, the speed of the gas in the inlet direction is consistent with the blade inlet angle. In Figure 10c, one can see that a small part of the annular flow path in the impeller is blocked, and a decrease in the flow rate results in a decrease in the meridional velocity of the impeller inlet, followed by an increase in its velocity incidence angle, and the gas shoots to the working surface, resulting in flow separation on the non-working surface. Due to the influence of the uneven airflow and the shape of the whole fan passage, there are always several passages which expand the separation zone firstly, and then form the separation cluster, extending to the whole blade passage. Therefore, from 12 o'clock to 5 o'clock, a large area of separation vortex appears in the blade path, blocking the blade path. In Figure 10b, a large number of flow separation vortexes appear on the pressure surface of

the blade between 12 o'clock and 4 o'clock, and a large part of the annular flow path in the impeller is blocked, which can result in a large-scale rotating stall. As the velocity steadily decreases in the non-working surface around the outlet of impeller, and the pressure gradually increases, the gradient of adverse pressure increases, making the boundary layer separation further expand. In other parts of the blade passage, due to uneven flow distribution, the flow rate in these passages has been reduced to a critical value, and the separation zone was seriously expanded to fill the whole blade passage. Meanwhile, in Figure 10a one can see that a large amount of vortex flow almost fills the internal flow of the impeller, which also indicates that the internal flow of the impeller is fully blocked by the separated vortex flow.

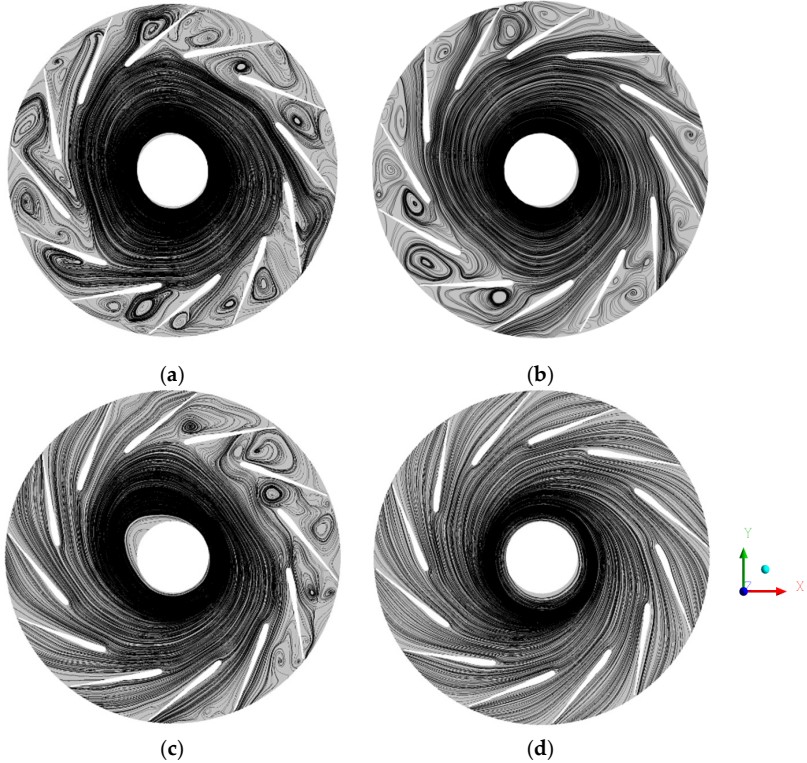

**Figure 10.** Streamline of impeller at z = 80 mm on the surface of revolution. (**a**) 0.15 $Q_d$, (**b**) 0.3 $Q_d$, (**c**) 0.5 $Q_d$, (**d**) $Q_d$.

In order to study the separation flow, the blade channel at 12 o 'clock in Figure 10a is partially enlarged and analyzed, as shown in Figure 11. In Figure 11, the leading edges of the two blades are marked as points 1 and 2, and the position of the suction surface near the exit is marked as point 3. At point 1, the impingement angle of impeller inlet velocity increases due to the decrease of flow rate at the condition of small flow rate, resulting in flow separation on the suction surface. At point 3, due to the small flow rate, the outlet velocity of blade decreases, the pressure increases, and the gradient of pressure pointing to the pressure surface is generated, leading to the expansion of the separation area. At the tail of the separation region, a separation vortex is formed inside the blade passage, and the rotation direction of the separation vortex is clockwise. This phenomenon is exactly the opposite of that of the impeller rotation direction. The air velocity decreases in the blade passage. In order to reach the equilibrium state, a torque balance impeller is required to act on the convective mass point. The expression of this torque is to produce a clockwise rotating separation vortex group. Squeezed by a separation vortex group and separation flow, the impingement angle of the impeller inlet tends to increase at point 2, so that the downstream blade passage is completely blocked by a large number of separated vortices, which explains that air had passed through a small number of blade channels.

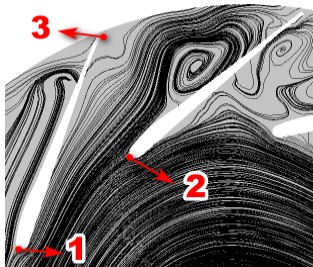

**Figure 11.** Magnified image of individual channel. Leading edges of the two blades are marked as points 1 and 2, and the position of the suction surface near the exit is marked as point 3.

The three-dimensional streamlines are used to reveal the instantaneous characteristics of the flow field and the direction of each particle. Figure 12 shows the three-dimensional streamline inside the volute. Illustrated in Figure 12, it can clearly be seen that the outlet velocity of the impeller and the outlet velocity of the volute at the design condition is greater than that of the other small flow conditions. At the design condition, the steady and smooth flow passes through impeller and volute, and no backflow occurs. However, at other small flow rates, a few backflows appear near the outlet area of the volute and an increasingly backflow gradually increases with the decrease of the flow. Figure 13 illustrates the three-dimensional streamline inside the impeller. As shown in Figure 13, it is observed that the inlet velocity of the impeller is high at the design condition, and the air smoothly enters into the blade passage. With the decrease of flow rate, the inlet velocity of the impeller not only decreases, but also the uneven flow distribution appears at different inlets of the blade. This is due to the flow separation and the shape of the flow channel, resulting from the decrease in the flow rate. At the $0.15Q_d$, there is no streamline in some blades, which indicates that the flow rate of these blades is very low, almost completely blocked, and the inside of these blades are full of a large amount of vortex flow.

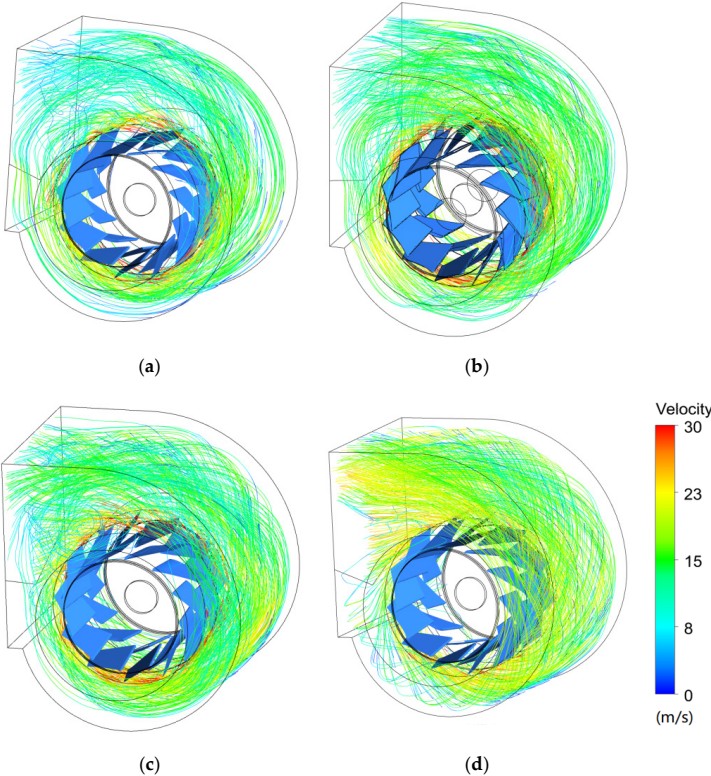

**Figure 12.** Three-dimensional streamline of impeller and volute. (**a**) 0.15 $Q_d$, (**b**) 0.3 $Q_d$, (**c**) 0.5 $Q_d$, (**d**) $Q_d$.

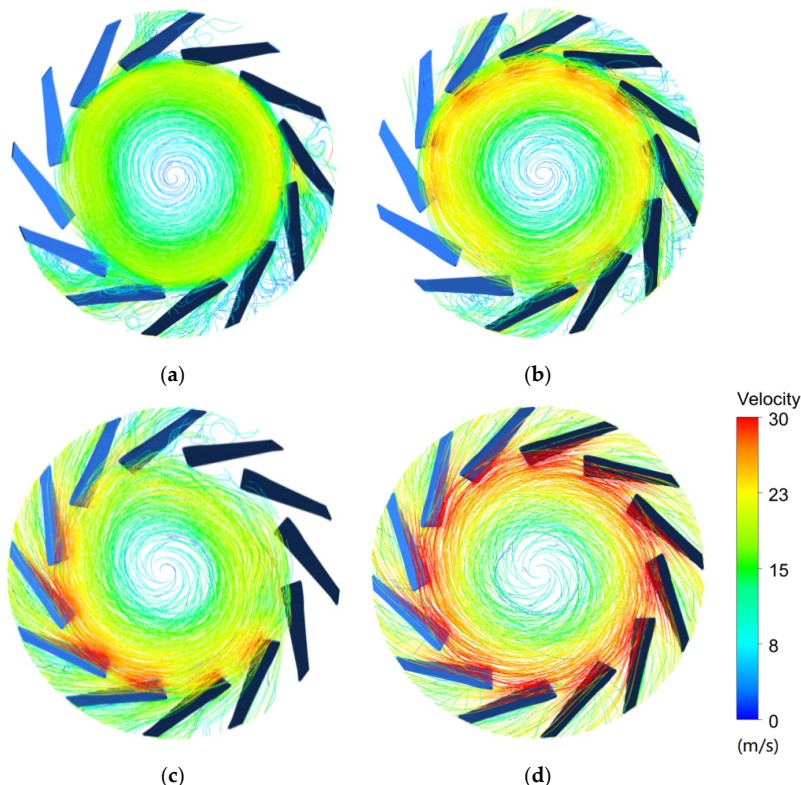

**Figure 13.** Three-dimensional streamline in the impeller. (**a**) 0.15 $Q_d$, (**b**) 0.3 $Q_d$, (**c**) 0.5 $Q_d$, (**d**) $Q_d$.

4.2.2. Secondary Flow Distribution

The secondary flow also conversely affects the main flow. In this subsection, the characteristics of secondary flow at small flow will be discussed and explored. Figure 14 shows the streamline of impeller and volute on the meridian surface. It can be obviously observed that a pair of vortex flow occurs on both sides in the volute at the design flow. These air flows from the impeller are influenced by the centrifugal inertia force on the volute wall, resulting in increased pressure on the volute wall. While the wall pressure becomes small on both sides of the volute, a pair of vortex flows are formed due to the pressure difference. It is not hard to imagine that the superposition of such a vortex flow and the main flow would result in a double spiral flow. This phenomenon can be observed at the designed flow rate in Figure 14d. This phenomenon is mainly due to the flow of the main stream decreases, and there is not enough energy to take the air away from the low-lying area. Meanwhile, it is also seen that vortices are not distributed near the border of the meridional surface. In the bottom right corner of Figure 14b, the above three vortices are observed. The decrease in potential energy causes the secondary vortex pair to converge to the central axis of the meridional plane. The regions of all vortices are also squeezed by other vortices, and their relative positions are constantly changing. When the flow rate drops to 0.3 $Q_d$, these vortices are also generated at the outlet of the impeller, which is caused by flow separation at the outlet due to the blocked blade passage. As shown in Figure 14a, it is further noted that a great deal of vortices derived from corner vortices and shroud covers appear, and the number and scope of vortices continue to expand at the work condition of 0.15 $Q_d$.

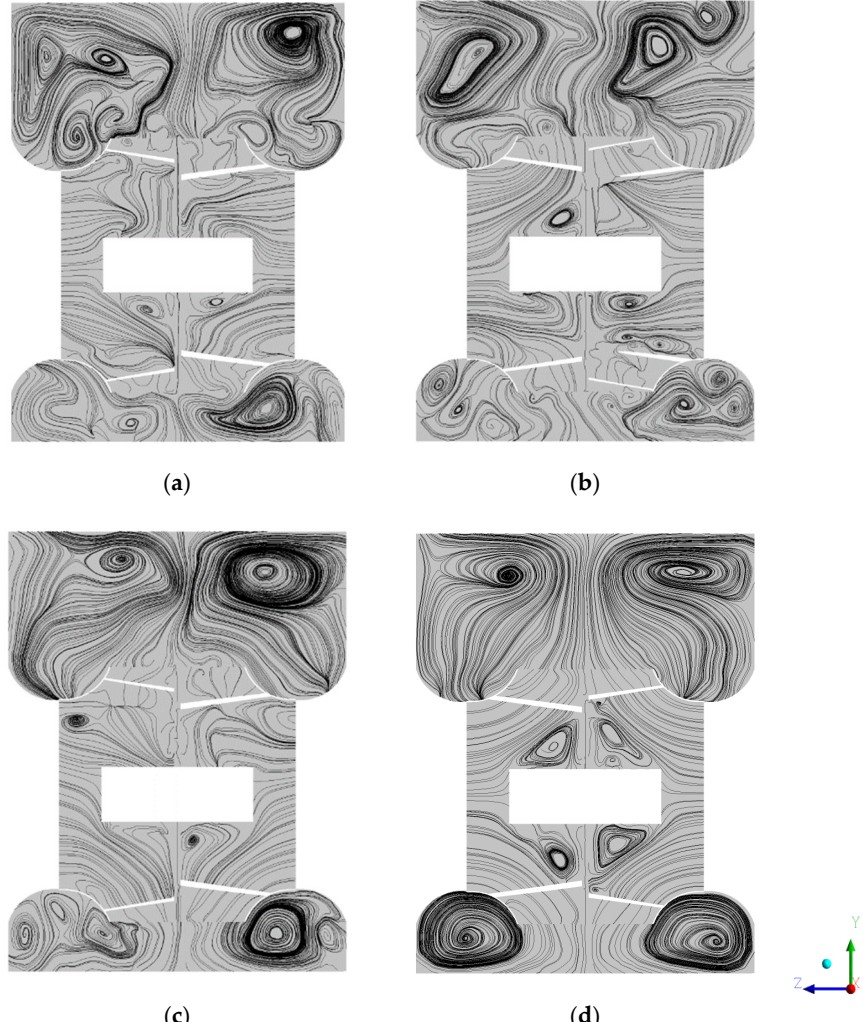

**Figure 14.** Streamline of impeller and volute on the meridian surface. (**a**) 0.15 $Q_d$, (**b**) 0.3 $Q_d$, (**c**) 0.5 $Q_d$, (**d**) $Q_d$.

Figure 15 shows the streamline of impeller on the circumferential surface. A torus is created at 0.5 times the blade length in the impeller, and the streamline is displayed on the tori, then the secondary flow distribution occurs in the blade passage. In this paper, the torus side view is selected, which can accurately observe the flow field in the three blade channels. In Figure 15, it is marked by 1, 2 and 3 respectively. Some researchers have pointed out that the non-uniform distribution of flows between the shroud and the wheel is the reason for the rise of pressure pulsation on the blade surface [15,16]. It is found that the flow separation and reverse flow not only occur in the blade channel, but secondary flows also appear in the blade channel. In Figure 15d, at the design flow rate, the air smoothly flows through the blade passage, and the fluctuation perpendicular to the mainstream is basically not observed. In Figure 15c, when the flow rate is 0.5 $Q_d$, the air in channel 2 and 3 fluctuates widely in the z-direction. In channel 1, the air is observed to gather from the shroud and the wheel to the middle of the blade, and two vortices are formed along opposite directions in this area. At the top of channel 1, facing the suction surface, a large amount of air pours in, creating a vortex. Although the secondary flow is different from that in the volute, this vortex is obviously not caused by centrifugal force around the flow, but by surface suction. This effect does not necessarily produce a pair of vortices, which are shown as transverse fluctuations in the 2 and 3 channels in Figure 15c, and as single vortices in Figure 15a 2 and 3 channels. This phenomenon indicates that the difference of pressure between the suction surface and the pressure surface and the potential flow in the blade channel dominate

the distribution characteristics of secondary flows in the blade channel. Therefore, according to the literature [16], the thickness of the secondary flow layer is expressed as:

$$\delta = f\left(\frac{\nu}{\upsilon}\right) \tag{10}$$

where $\nu$ is the kinematic viscosity coefficient $\upsilon$ is the velocity of air.

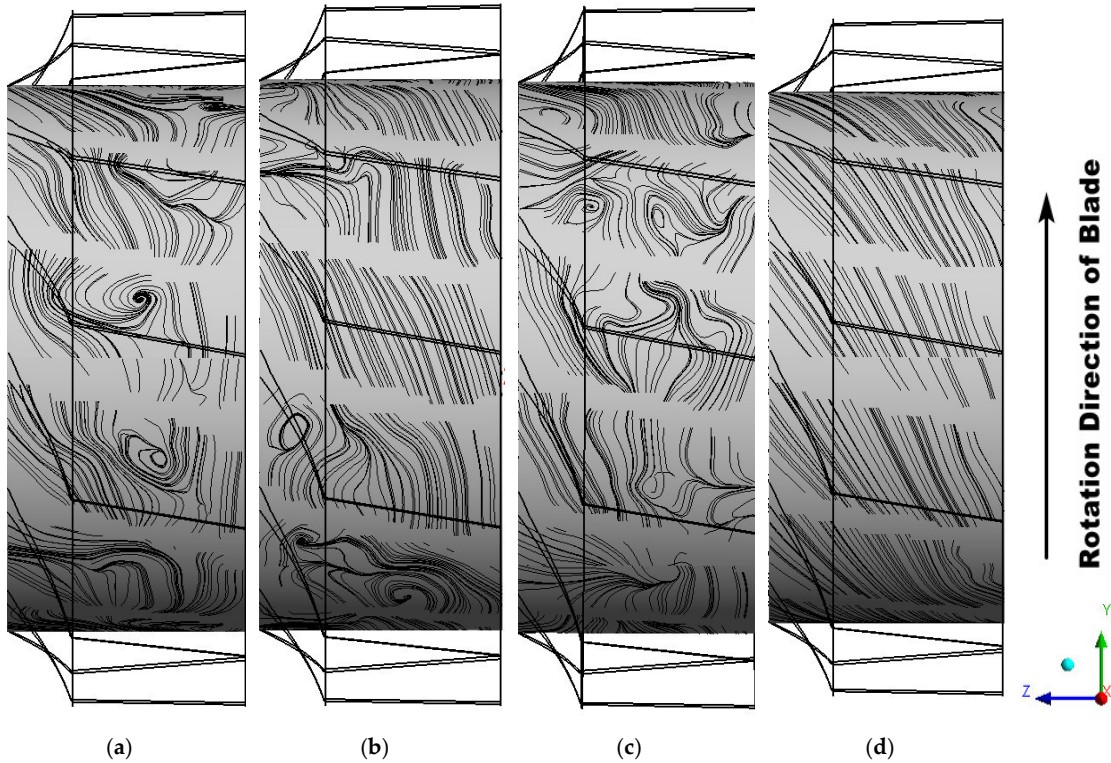

**Figure 15.** Streamline of impeller on the circumferential surface. (**a**) 0.15 $Q_d$, (**b**) 0.3 $Q_d$, (**c**) 0.5 $Q_d$, (**d**) $Q_d$.

### 4.3. Internal Unsteady Complex Flow with Time Evolution

This section mainly discusses the unsteady characteristics of internal flow with time evolution. Considering the unsteady flow characteristics of the fan, the time step of the numerical simulation is set to 0.0002381 s, which means that the impeller rotates 2° in each time step. In this study, the Courant number is equal to 200. Two parts are classified as the distribution of the flow field after the steady state and the initial stage of the flow field development. LES is implemented to capture the large-scale flow details of the unsteady flow. The surge of the fan is regarded as a self-excited cyclically nonlinear phenomenon, and at least four different surge categories are distinguished with respect to unsteady flow and pressure fluctuations.

In order to capture the complex unsteady turbulent flow of impeller in low flow rate, the flow rate is implemented at 0.15 $Q_d$. Figure 16 illustrates that the streamline on the rotary surface of the impeller rotates one turn at 0.15 $Q_d$. As shown in Figure 16, one can obviously see that three channels maintain circulation; others are blocked by separating vortices comparing the flow chart of different time. Interestingly, a large number of separation vortex occur in the pressure surface, suction surface of blade and internal passage of impeller, and gradually develops with rotating impeller. The clock channel is executed to capture the unsteady flow of an impeller at 0.25 turn, 0.5 turn, 0.75 turn and 1 turn. To observe the 11 o'clock channel in Figure 16a, when the impeller rotates a quarter turn (see in Figure 16b), it is clearly seen that the channel is blocked by the separation flow, while the original flow only develops to the direction of 9 o'clock. This indicates that the separated vortices gradually move

in the opposite direction to the impeller. Compared with the four pictures in Figure 16, the rotating speed of the separating vortices is half that of the impeller. In Section 4.1, the separation in the blade channel has been discussed, and it is found that the generation of the separation vortex in the blade channel is to balance the torque generated by the rotation of the impeller. It is further demonstrated that the separation flow vortices begin to downstream with time evolution, which is mainly due to the separation vortex rotation and impeller rotation direction is opposite. Meanwhile, the separation flow vortices will affect the downstream channel outlet. Similarly, the separation flow blocks the passage, causing the air to fail to pass smoothly, indirectly increasing the inlet angle of attack of the downstream passage. The migration of the separated vortex is fully captured from one channel to another.

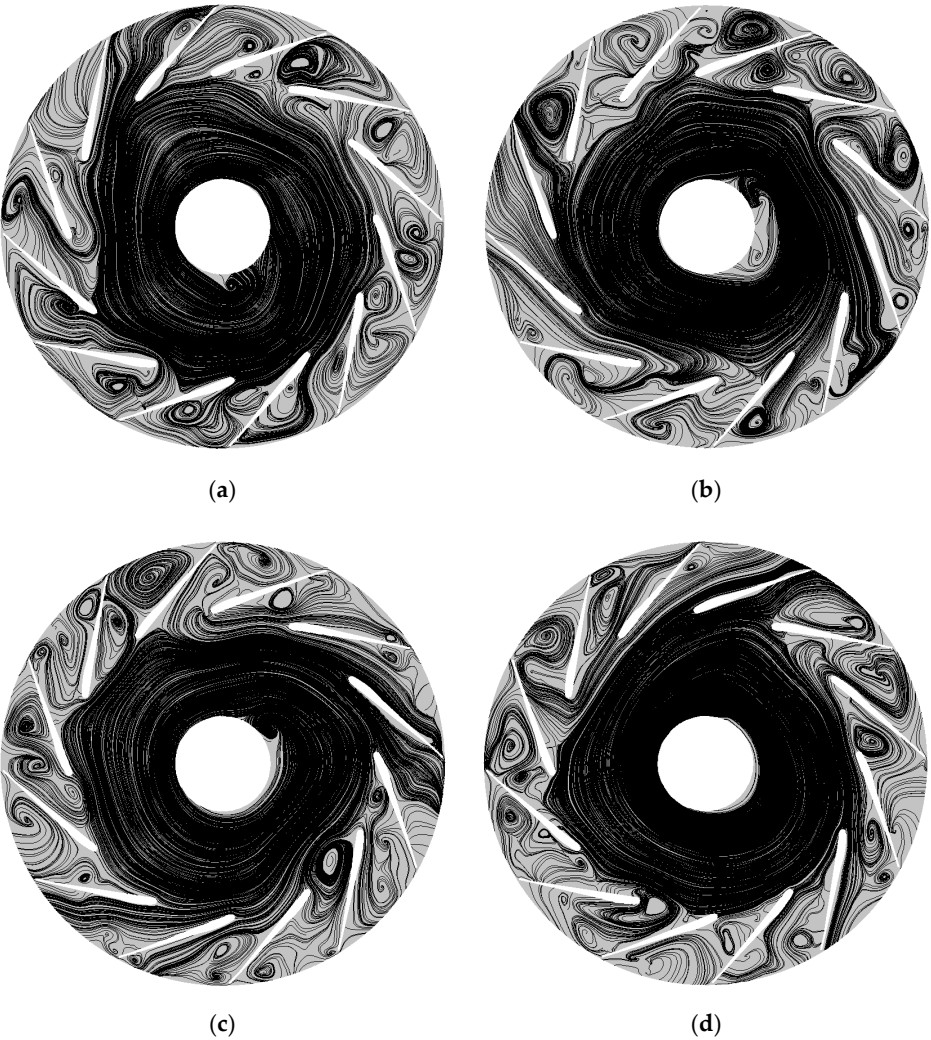

(a)  (b)

(c)  (d)

**Figure 16.** Streamline on the rotary surface of the impeller rotates one turn at 0.15 $Q_d$. (**a**) 0.25 turn, (**b**) 0.5 turn, (**c**) 0.75 turn, (**d**) 1 turn.

In order to further explain the above unsteady physical phenomenon, the rotary surface of the impeller changed is used at 0.15 $Q_d$ with time evolution. Figure 17 shows that the streamline on the rotary surface of the impeller changed with time evolution at 0.15 $Q_d$. Figure 17 shows a transient image every 6°, which can accurately show the change of the flow field in the blade channel and the development of the separation vortex. In most blocked channels, the suction surface near the inlet produces a separation vortex due to flow separation. It is obtained that the inlet of the channel 11 just forms a small vortex in Figure 17a and adheres to the suction surface, then the scope of the vortex increases in Figure 17b. The vortices are taken away by the mainstream and move to the next blade

in Figure 17c. Until the vortex is destroyed by the mainstream in Figure 17d–f, the import forms a wide range of flow separation. In most channels (e.g., channel 3), vortices form that block the channels. As observed in the development of streamline in channel 1, one can see that the inlet of the blade channel gradually forms a suitable angle of attack, the separation vortex in the blade channel will be squeezed by the flow gas and finally adhere to the pressure surface. As the flow of air increases, this separation vortex will disappear. As shown in Figures 16 and 17, some insights are obtained that show how, due to a large number of separated vortex filling with internal passage of impeller, the rotating speed of unsteady separated vortex increasingly decreases. Nevertheless, the impeller still rotates at a constant speed. Thus, these separated vortices intensely act on the pressure surface of the blade. The interaction between the separated vortices and surface of blade will cause increasingly small-scale eddies. Finally, the internal flow of the impeller is the non-axisymmetric flow, which is fully blocked by the increasing vortex flow. These insights can clearly explain the rotating stall and surge physical phenomenon of the centrifugal fan.

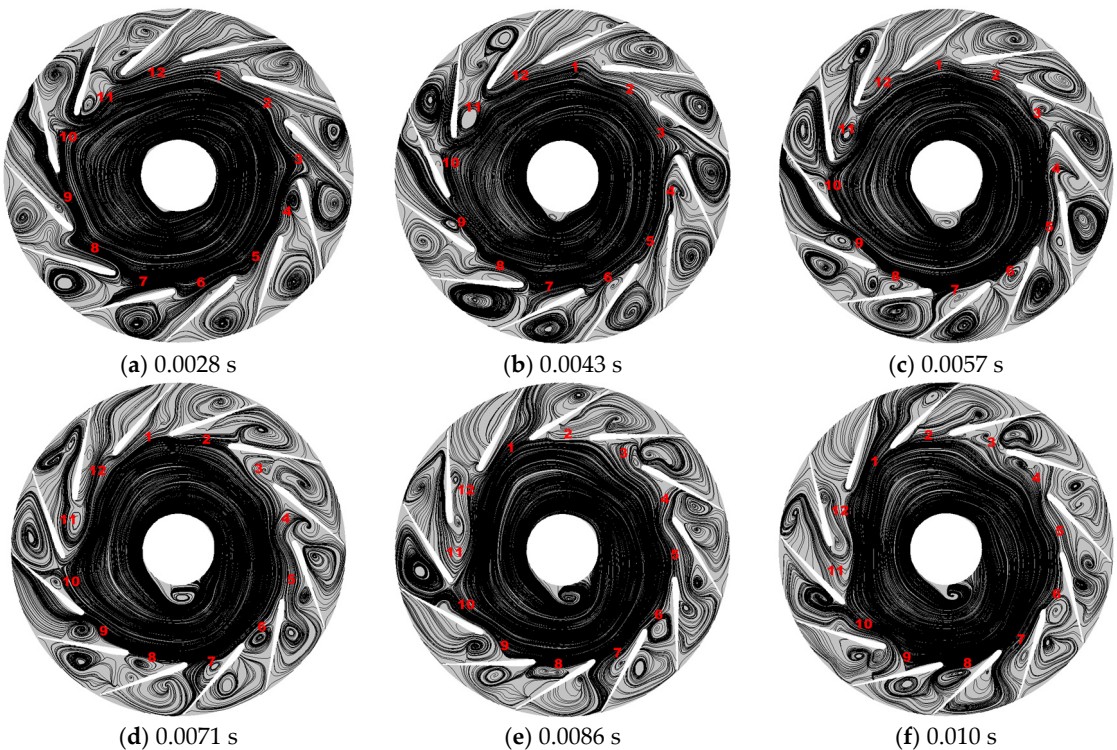

| (**a**) 0.0028 s | (**b**) 0.0043 s | (**c**) 0.0057 s |
| (**d**) 0.0071 s | (**e**) 0.0086 s | (**f**) 0.010 s |

**Figure 17.** Streamline on the rotary surface of the impeller changed with time evolution at $0.15Q_d$.

### 4.4. Fluctuation and Its Spectral Analysis at Low Flow Rates

In general, the volute tongue and central position of impeller channel are two important points to capture the unsteady characteristic of pressure and velocity, which is mainly regarded as an important criterion to capture the stall and surge phenomenon of the centrifugal fan [14]. Bianchi et al. [14] argued that the rotating stall is not a single phenomenon, but rather four obvious phenomena of part span, full span, small scale and large scale. The pulsation of the flow field in the impeller and the volute will be investigated. Two monitoring points as shown in Figure 18 are selected, one is located in the blade channel inside the impeller, and the other is located at the volute tongue.

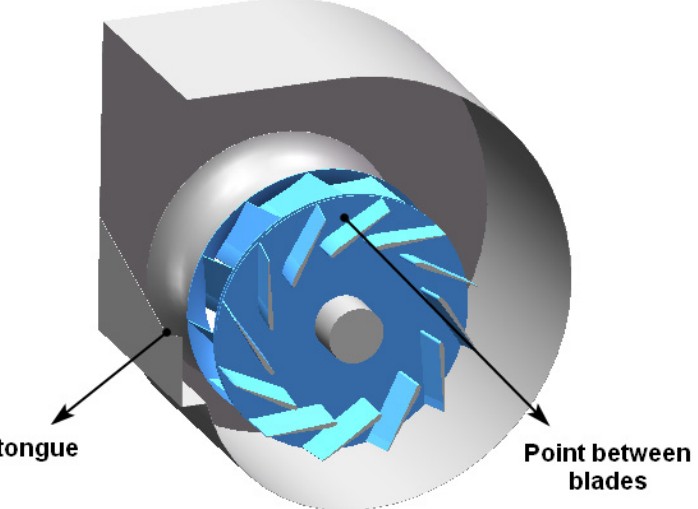

**Figure 18.** Monitory points.

### 4.4.1. Pulsation Time Domain Analysis of Pressure and Velocity

Figure 19 shows the pressure fluctuations at the tongue under various working conditions (0.15 $Q_d$, 0.3 $Q_d$, 0.5 $Q_d$ and $Q_d$). Plotted in Figure 19, it is obviously observed that no matter how to change the working condition, pressure at the volute tongue is not a fixed value, within the scope of a constant volatility. When the flow rate is $Q_d$, the amplitude of pressure fluctuations is relatively small at the tongue. It is also obtained that, with decreasing the flow rate, the amplitude of pressure fluctuations increases and the value of pressure on the whole gradually relatively decreases from 0.3 $Q_d$ to $Q_d$, which indicates that the unsteady characteristics of gas acting periodically on the tongue gradually increases with the decrease of flow rate in this range. Nevertheless, at the working condition of 0.15 $Q_d$, It is found that the value of pressure sharply increases on the whole, which indicates that a large number of separated vortices fill the entire impeller passage, as shown in Figures 16 and 17. Figure 20 describes the velocity fluctuations at the tongue at various flow rate. As shown in Figure 20, it is clearly observed that the amplitude of velocity fluctuations gradually increases with the decrease of flow rate, which further demonstrates that the unsteady characteristics of gas acting periodically on the tongue gradually increase.

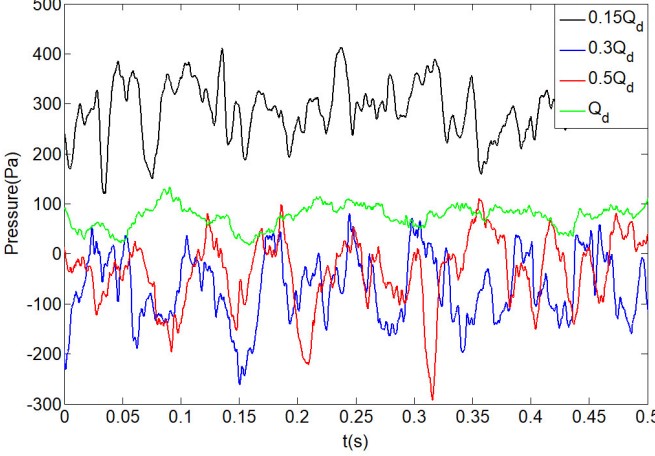

**Figure 19.** Pressure fluctuations for the tongue at various working conditions.

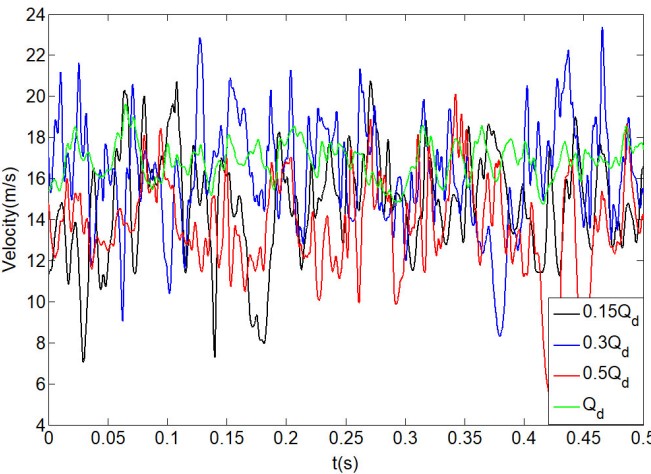

**Figure 20.** Velocity fluctuations for the tongue at various working conditions.

Figure 21 shows the pressure fluctuations of the blade inside at various working conditions. We can see that at the design flow rate, the air pressure in the blade passage fluctuates steadily, and the fluctuating range is relatively stable, and stable fluctuation cycles can be captured. When the flow rate decreases, the pressure fluctuation becomes chaotic and the wide range of fluctuation gradually increases. However, it is also demonstrated that the value of pressure sharply increases on the whole at the working condition of 0.15 $Q_d$, which further indicates that a large number of separated vortices seriously fill the entire impeller passage. It is further indicated that, at the working condition of 0.15 $Q_d$, the air in the passage is not only periodically pushed by the blade, but also, due to the decrease of flow rate, the flow shock or flow separation vortex occurs in the passage or other basins, which makes the pressure fluctuation in the passage become chaotic. Figure 22 illustrates the velocity fluctuations of the blade inside at various working conditions. As illustrated in Figure 22, it is clearly seen that the amplitude of velocity fluctuations gradually increases with the decrease of flow rate at a range flow rate of $Q_d$ to 0.3 $Q_d$, which further indicates that the unsteady characteristics of flow in the blade inside gradually increases with the decrease of flow rate in this range. Nevertheless, at the working condition of 0.15 Q, it is found that the amplitude of velocity fluctuations in the blade channel become weakened compared to that of 0.3 $Q_d$, which implies that the entire impeller passage filled a large number of separated vortices, and the unsteady characteristics of flow in the blade inside also become weakened. In other words, the centrifugal fan cannot run completely, and might have maintained the rotating stall and surge.

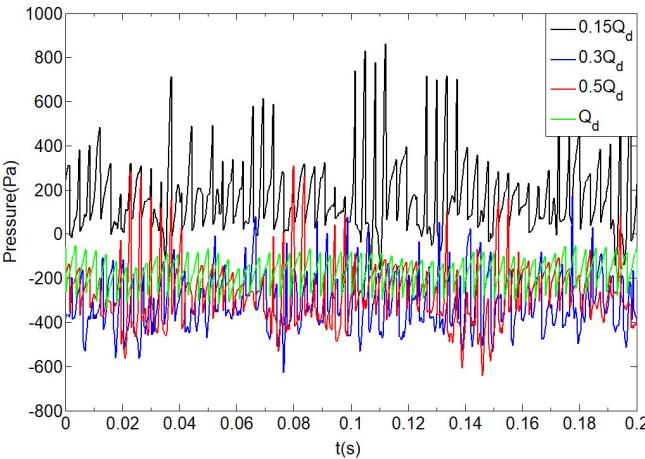

**Figure 21.** Pressure fluctuations of the blade inside at various working conditions.

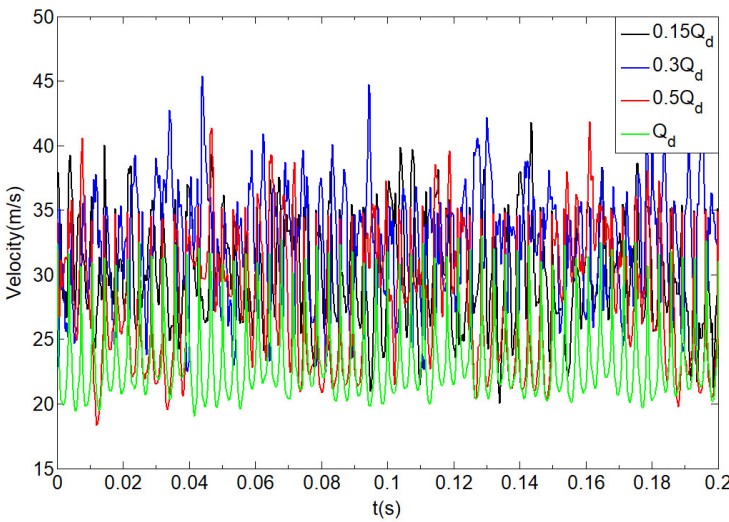

**Figure 22.** Velocity fluctuations of the blade inside at various working conditions.

4.4.2. Fast Fourier Transform Analysis of Pressure and Velocity Fluctuations

In general, the fast Fourier transform (FFT) is a very effective tool to reveal the insights of these unsteady flow fluctuations since it is the most widespread and dependable method for industrial application. A large number of disturbance flow characteristics are captured by the interaction between rotating parts and stationary parts in the fan, and even the interaction between the air and the internal structure of the fan. This disturbance is mainly responsible for the fluctuations of pressure and velocity of the flow field, and the flow will transfer this disturbance to the impeller and volute, which is also responsible for mechanical vibration and noise. To analyze the frequency of pressure and velocity changes inside the centrifugal fan, the time-domain signals of pressure and velocity fluctuations in Figures 19–22 are transformed into the frequency domain signals by the FFT in Figures 23 and 24. The peak in the frequency domain signal corresponds to the frequency of the pressure pulsation.

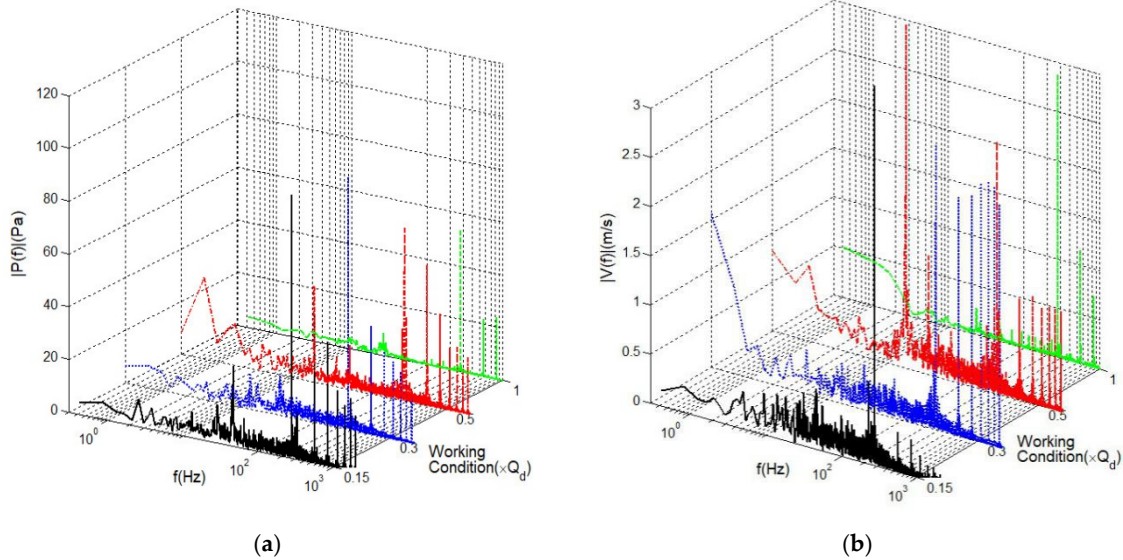

(a) (b)

**Figure 23.** Fluctuations for the tongue at various working condition. (**a**) Amplitude spectrum of pressure fluctuations, (**b**) Amplitude spectrum of velocity fluctuations.

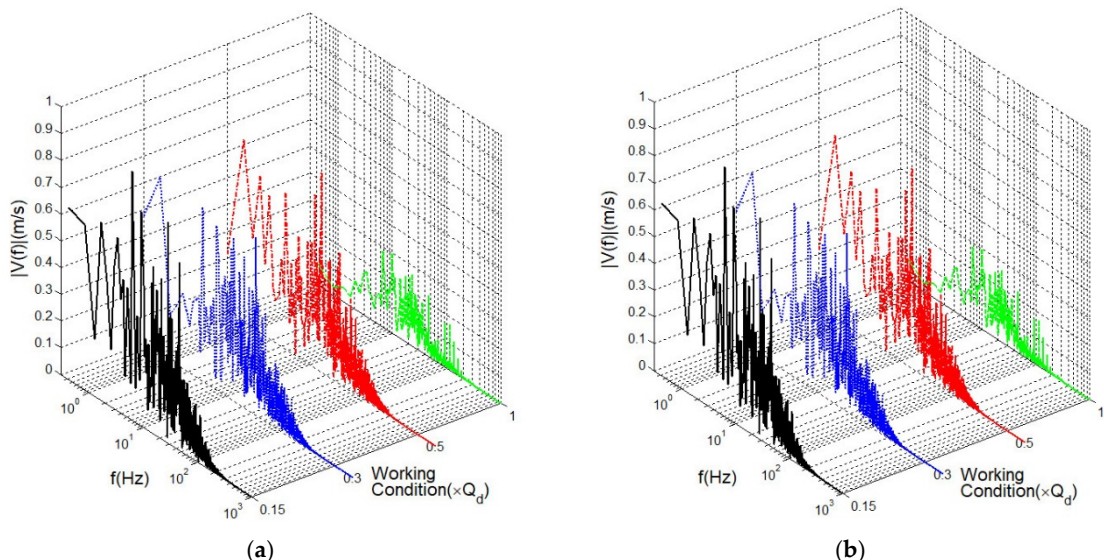

**Figure 24.** Fluctuations of the blade inside at various working condition. (**a**) Amplitude spectrum of pressure fluctuations, (**b**) Amplitude spectrum of velocity fluctuations.

In turbomachinery movement, because the impeller blade regularly pushes the air, this will cause air pulsation. The theoretical calculation formula of the pulsation frequency of the centrifugal fan is [18]:

$$f_i = \frac{nZ}{60} i \tag{11}$$

where $i$ is order of harmonics, $i = 1, 2, 3 \ldots$ , n is the speed of impeller, r/min, and Z denotes the number of blades.

Figure 23 shows the amplitude spectrum of fluctuations for the tongue at various working conditions. As illustrated in Figure 23a, it is clearly observed that at the design flow rate, the three pulse peaks of the amplitude spectrum of pressure fluctuations appear in a range of 100 Hz to 1000 Hz. At the working condition of 0.5 $Q_d$, it is obtained that the first strong peak occurs at 50 Hz and a great deal of strong peaks appear in a range of 200 Hz to 1000 Hz. However, it is also found that the amplitude spectrum value of the strongest peak of pressure fluctuations gradually increases with the decrease of flow rate, and a large number of peaks appear irregularly in the proper range of frequency bands at the work conditions of 0.5 $Q_d$, 0.3 $Q_d$ and 0.15 $Q_d$. As described in Figure 23b, one can see that the three strong peaks of the amplitude spectrum of velocity fluctuations mainly occur in a range of 100 Hz to 1000 Hz. Nevertheless, it is obtained that for the working condition of 0.5 $Q_d$ the first strong peak of amplitude spectrum of velocity fluctuations appears at 20 Hz and a great deal of strong peaks appear in a range of 200 Hz to 1000 Hz. For the working condition of 0.3 $Q_d$, the first strong peak of amplitude spectrum of velocity fluctuations appears at 200 Hz and many stronger peaks appear in a range of 200 Hz to 1000 Hz, which implies that the interaction between the volute tongue and the internal structure of the centrifugal fan increasingly strengthens in a range of $Q_d$ to 0.3 $Q_d$ with the decrease of flow rate. Interestingly, it is surprisingly found that a strong peak appears at 200 Hz for the working condition of 0.15 $Q_d$. This mainly indicates that the unsteady characteristics of acting on the volute tongue gradually increases in a range of $Q_d$ to 0.3 $Q_d$ with the decrease of flow rate and the unsteady characteristics acting on the volute tongue is weakened at the working condition of 0.15 $Q_d$ due to the appearance of a large amount of reflux flow near the volute tongue, which further demonstrates the above qualitative analysis of Figure 9.

Figure 24 shows the amplitude spectrum of fluctuations for pressure and velocity inside the blade with the decrease of flow rate. As shown in Figure 24a, one can see that when the frequency is the range of 10 to 100 Hz, the amplitude spectrum value of strongest peak of pressure fluctuations gradually increases in a range of $Q_d$ to 0.3 $Q_d$ with the decrease of flow rate. Nevertheless, the amplitude

spectrum value of pressure fluctuations decreases compared with that of the working conditions of 0.5 $Q_d$ and 0.3 $Q_d$. This mainly indicates that as the vortex shedding period around the blade increases, the spectral transport of energy also increases, and the turbulent structure of noise source in the blade channel caused by aerodynamics enhance the working conditions of 0.5 $Q_d$ and 0.3 $Q_d$. However, the above-related characteristics in the blade inside is weakened at the working condition of 0.15 $Q_d$ due to the blocked internal flow of impeller by the separated vortex flow, which further demonstrates the above qualitative analysis of a series of steady flow.

According to the previous studies of classifying the rotating stall and surge [14], Figures 21–24 further indicate that the working condition of the centrifugal fan might have been the rotating stall of large scale at the flow rate of 0.3 $Q_d$ and the working condition of centrifugal fan was the severe version of classical surge at the flow rate of 0.15 $Q_d$.

## 5. Conclusions

The steady and unsteady characteristics of the flow are studied at low flow-rate conditions. Some conclusions obtained in this paper are as follows:

In first, it is demonstrated that the numerical simulation results of aerodynamic performance are in excellent agreement with that of experimental tests.

Moreover, the distribution of secondary flows are captured on the meridional plane with the decrease of flow rate, small scale vortices are generated and formed in the region, a great deal of vortices derived from corner vortices and shroud covers appear, and the number and scope of vortices continue to expand at the working condition of 0.15 $Q_d$.

In addition, with the decrease of flow rate, the unsteady separated vortex gradually appears and increases. Due to the gradual asynchrony between the separated vortices and rotating impeller, the interaction between the separated vortices and surface of the blade will increasingly enhance. The internal flow of the impeller is gradually blocked by the increasing vortex flow.

Furthermore, the fluctuations, and the spectrum analysis of the fluctuations of pressure and velocity indicate that the unsteady characteristics of acting on the volute tongue gradually increases in a range of $Q_d$ to 0.3 $Q_d$ with the decrease of flow rate and the unsteady characteristics acting on the volute tongue is weakened at the working condition of 0.15 $Q_d$ due to the blocked internal flow of the impeller by the separated vortex flow. The vortex shedding period around the blade increases, the spectral transport of energy also increases, and the turbulent structure of noise source in the blade channel caused by aerodynamics are enhanced in the working conditions of 0.5 $Q_d$ and 0.3 $Q_d$. However, the above-related characteristics inside the blade are weakened at the working condition of 0.15 $Q_d$.

Finally, the steady flow and unsteady flow further reveal that one working condition of the centrifugal fan was the large scale rotating stall at the flow rate of 0.3 $Q_d$, and the other working condition of the centrifugal fan was the severe surge at the flow rate of 0.15 $Q_d$. These insights can clearly explain the rotating stall and surge physical phenomenon of the centrifugal fan, which can be extensively applied to help many kinds of engineering designs for centrifugal fans.

**Author Contributions:** Investigation, Writing, B.Z.; Methodology, X.H.; Conceptualization, Writing, H.Y.; Supervision, Z.Z.; Conceptualization, Validation, Y.W.; Validation, Formal analysis, Y.Z. All authors have read and agreed to the published version of the manuscript.

**Funding:** This work was supported by National Natural Science Foundation of China (51906223, 11872337, and 91841303), Fundamental Research Funds of Zhejiang Sci-Tech University (2019Y004), Key Research and Development Program of Zhejiang Province (2020C04011) and Public Projects of Zhejiang Province (LGG20E060001). The supports are gratefully acknowledged.

**Conflicts of Interest:** The authors declare no conflict of interest.

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
