# Peer review of "Unsteady Flow Characteristics of Rotating Stall and Surging in a Backward Centrifugal Fan at Low Flow-Rate Conditions"

_processes, doi:10.3390/pr8070872_

Round 1

Reviewer 1 Report

Review of Zhou et al. 'Unsteady Flow Characteristics of Rotating Stall and  Surging in a Backward Centrifugal Fan at Low Flow-rate Condition'

Summary:

The paper by Zhou et al. examines the steady and unsteady flow characteristics of a centrifugal fan comprising of a double inlet subjected to several flow rate conditions. At certain reduced flow rate conditions, they find that the 'fluctuations' in velocity and pressure amplitude tends to increase and more specifically at 0.15Q, suggesting the inherent nature of the flow being truly unsteady. The results appear to be sensible, and in my opinion, it is a good, comprehensive paper that can be published in Processes.

I would suggest a few minor modifications to the paper before its publication, as follows:

1) Please change equation 10 to a proper format.

2) For the readers to understand it a bit better, could you change the Figure captions of Fig.4? Instead of saying the total grid number (thousand): Use 103 or re-scale the graph and say 106 it is not clear what thousand means in this context.

3) I have a feeling that this was executed using possibly, the CFX. If you have used CFX or Fluent or any commercial code and that it is not a part of your own code- please acknowledge what code you have used perhaps say this at the start of Section 3. In case you have used an in-house code; just say in-house etc.

4) Either the time-step or an equivalent CFL number could be useful for the readers. Therefore, please provide that.

5) On page 8: 'obviously, ,' remove double commas.

6) In the abstract section,' These insights clearly explain the rotating stall and surge physical phenomenon of centrifugal fan'. Instead, say what it explains- it looks like there is a loose end you need to tie. For example, you could say, these insights clearly explain the unsteady nature of the rotating stall and surge phenomenon exhibited by the double inlet backward centrifugal fan.

7) Overall, English should be improved significantly- although the technical content appears to be good. For example, on page 25:

In final, these steady flow and unsteady flow further reveal that at the flow rate of 0.3Q, the working condition of centrifugal fan can have been the rotating stall of large scale and the working condition of centrifugal fan can have been the severe version of classical surge at the flow rate of 0.15Q. These insights can clearly explain the rotating stall and surge physical phenomenon of centrifugal fan, which can be extensively applied to help many kinds of engineering designs for centrifugal fan.

Reviewer 2 Report

The subject material is presented in a comprehensive manner, however, authors could improve some aspects that I suggest.

The present work is contextualized with current or recent bibliography.

Correct some formatting problems: some points in the text a word and identification of the reference without space between them; formatting reference 31...

Some figures subtitles with lack of information in order to make their visualization and available information easy to understand (for example: figures 6, 7, 8, 9, 10,  ... explained what (a), (b),... represents or what can be seem in each other); units (verify the units used for the flow rate m3/s or m3/h ?); line 86: “3” as power. Check or rewrite equation 10 (line 372).

Along the reading of the text, it seems to be necessary a better clarification of what the authors consider “steady and unsteady flow characteristics” as mentioned in the abstract. The title, for example, only refers “unsteady flow characteristics...”, in some parts of the text (line72-73 Introduction) authors refer that “... study is mainly focused on the steady flow properties and the unsteady flow mechanism...” so, and characteristics? Are they properties or mechanism?... and section 4.2 is about “steady flow field distribution characteristics” (like pressure, velocity, ...), and 4.3 about “... unsteady characteristics of ... with time evolution.”

In the Abstract the letter “Q” is referred; what is its meaning? Later on in the paper, the letter “Q” appears representing “the design condition (Q) of...”, but it also represents the flow rate?! Clarify.

Abstract: it seems that is missing a value for Q (1?). Explain the meaning of Q (what Q represents (flow rate?), and units if necessary) (see previous comment).

Introduction: the ideas in the first paragraph of the Introduction section, seems to appear like "out of the blue" without any connection between then. The ideas do not flow. For example, authors start to identify the importance of centrifugal fans in the actual moment and then start to talk about technical requirement without, not only, by making a contextualization with references, but also without any guiding principle.

The identified works, from 8 to 13, are only enumerated without mentioned the importance of the obtained results in each regarding the changing/influence of the identified parameters (like authors mentioned regarding work [16]). Only later on authors explained this, however, this could be better articulated.

Section 3: line 97, CFD is more like a set of numerical methods to solve numerically the equations that describe the phenomenon. So, identify the numerical software that was used as an effective tool...

Section 4: lines 200-..., “at the designed working condition of 0.3 times (0.3Q)...” explain better the meaning of 0.3 times (0.3Q), and similar comment whenever this format appears in the text.

Also, whenever authors identify the clock hours, state that this corresponds to the position and in order to facilitate the reading and identification of the point under analysis. Do not identify only the clock hours...

Conclusions, authors must identify the conditions that were considered in the simulations since the results obtained were verified under a specific set of conditions and parameters.

And better reinforce the added value of the results presented in the context identified at the beginning of the paper.
